# Inborn Errors of Immunity Causing Pediatric Susceptibility to Fungal Diseases

**DOI:** 10.3390/jof9020149

**Published:** 2023-01-22

**Authors:** Peter Olbrich, Donald C. Vinh

**Affiliations:** 1Pediatric Infectious Diseases, Rheumatology and Immunology Unit, Hospital Universitario, Virgen del Rocío, 41013 Sevilla, Spain; 2Instituto de Biomedicina de Sevilla, IBiS/Universidad de Sevilla/CSIC, Red de Investigación Traslacional en Infectología Pediátrica RITIP, 41013 Sevilla, Spain; 3Departamento de Farmacología, Pediatría y Radiología, Facultad de Medicina, Universidad de Sevilla, 41009 Sevilla, Spain; 4Division of Infectious Diseases (Department of Medicine, McGill University Health Centre), Montreal, QC H4A 3J1, Canada; 5Division of Medical Microbiology, Division of Molecular Genetics-Immunology (Department of Op-tiLab, McGill University Health Centre), Montreal, QC H4A 3J1, Canada; 6Centre of Excellence for Genetic Research in Infection and Immunity, Research Institute-McGill University Health Centre, Montreal, QC H4A 3J1, Canada

**Keywords:** inborn errors of immunity, primary immunodeficiency, fungi, mycoses, invasive fungal disease, superficial fungal disease

## Abstract

Inborn errors of immunity are a heterogeneous group of genetically determined disorders that compromise the immune system, predisposing patients to infections, autoinflammatory/autoimmunity syndromes, atopy/allergies, lymphoproliferative disorders, and/or malignancies. An emerging manifestation is susceptibility to fungal disease, caused by yeasts or moulds, in a superficial or invasive fashion. In this review, we describe recent advances in the field of inborn errors of immunity associated with increased susceptibility to fungal disease.

## 1. Introduction

The inborn errors of immunity (IEI), also known as primary immunodeficiency disorders (PIDD), are a heterogeneous group of now ~485 disorders caused by monogenic germline mutations, that increase susceptibility to infections, autoimmunity, autoinflammatory disorders, atopy, bone marrow failure, and/or malignancy [1]. Among those that predispose to infections, the microbial spectrum can be broad (e.g., from several kingdoms) or restricted (e.g., to a single kingdom or even a single species); the corresponding infectious diseases can be recurrent, recalcitrant to therapy, and/or life-threatening. In addition, over the last decade, IEI that are saliently marked by fungal disease have been increasingly recognized. The development of fungal disease is frequently associated with an immunocompromised state. Often, this state is apparent clinically, for example, from medications/chemotherapy. In the absence of an exogenous source of immunosuppression, investigations for an inherent (i.e., genetic) cause should be pursued. Further, with the advent of genomic technologies and state-of-the-art immunological and molecular biological methods, IEI are defined molecularly and functionally, attributing a causal gene defect to the corresponding phenotype. The intersection of mycology with IEI has shed light on immunological pathways fundamental for human host defenses against fungi. The aim of this review is to provide physicians who encounter patients with unexplained fungal disease with an up-to-date reference for further genetic and immunologic investigations. The IEI discussed in this review is summarized in the Table 1.

## 2. Methods

A systematic search of the literature was performed in December 2022 to identify all publications reporting fungal disease in patients with IEI. Five databases were used: PubMed, Google Scholar, Web of Science, EMBASE, and Scopus. There was no restricted period of time; all articles dating back to the oldest one listed in the available database were identified. The literature search was performed using key words (and their derivatives) relevant to the fungi discussed in this review, including “fungi” (“fungal”), “yeast”, “mould” (“mold”), “mycosis” (“mycoses”), “Candida” (and “candidiasis”, “Torulopsis”, species names), “Aspergillus” (and “aspergillosis”, “Neosartorya”, species names, “thermally-dimorphic”, “endemic”, “Histoplasma” (and “histoplasmosis”, species names), “Blastomyces” (and “blastomycosis”, species names), “Coccidioides” (and “coccidiodomycosis”, “valley fever”, “San Joaquin valley fever”, species names), “Emergomyces” (and “emergomycosis”, species names), “Paracoccidioides” (and “paracoccidioidomycosis”, species names), “Mucor” (and “mucormycosis”, species names), “Rhizopus” (and species names), other fungal genera/species, in association with “inborn errors of immunity” (and “IEI”), “primary immunodeficiency” (and “PID”, “PIDD”), “gene” (and “genetic”, “mutation”, “variant”, “heterozygous”, “homozygous”, “loss of function”, “LOF”, “gain of function”, “GOF”, “dominant negative”, “negative dominance”, “haploinsufficiency”, “haploinsufficient”), “deficiency”, “immunodeficiency” (and “immune deficiency”), or individual IEI as listed in the “2022 IUIS Phenotypical classification for Human Inborn Errors of Immunity” [2]. The identified titles and abstracts were screened, and the full texts of suitable articles were reviewed.

The articles fulfilling the inclusion criteria were selected from the full-text articles. In order to be eligible for inclusion; a fungal disease must have been reported in patients from studies and case reports. Further, the IEI of patients must have been clinically specified and diagnosed by genetics. The relevant reviews and large case series were reviewed to identify potentially eligible original studies or unpublished data. We included all types of publications (articles, reviews, editorials, letters, and correspondences) written in English or French, and all references cited in these publications were also analyzed. Finally, where reports were identified that did not meet inclusion criteria, we have nonetheless included them with references, indicating that their contribution to the topic of this manuscript may be uncertain.

### 2.1. Candidiasis

*Candida* spp. can be found colonizing the skin, oral mucosa, and/or the gastrointestinal and genitourinary tracts in healthy individuals. Most cases of subsequent disease usually emerge from this endogenous microbiota [3,4]. Although there are now more than 200 species of *Candida* identified, only a relatively small and yet not clearly defined number (~10–20) have been reported in the context of disease in children and adults [5]. Historically, *C. albicans* has been, by far, the most commonly isolated species, although the rates of non-*albicans* species have variably increased globally, probably due to modifications in prophylactic approaches as well as changes in the characteristics of the most vulnerable patient populations, such as preterm neonates and immunosuppressed children [6,7,8]. Overall, *Candida* infections in children can be divided into two main disease presentations: chronic mucocutaneous candidiasis, which can manifest as oropharyngeal candidiasis (OPC, also known as “thrush”), esophagitis, diaper dermatitis, onychomycosis, and/or vulvovaginitis, and invasive candidiasis (IC).

#### 2.1.1. Chronic Mucocutaneous Candidiasis (CMC)

Mucocutaneous infections, such as oral thrush or diaper dermatitis, are relatively common in pediatrics. These infections are mostly found in the context of concomitant antibiotic treatment, topical or systemic corticosteroid therapy, or breakdown of the local skin barrier [8]. When mucocutaneous candidiasis is persistent or recurrent and the aforementioned risk factors are absent, the condition is often referred to as “chronic mucocutaneous candidiasis” (CMC). Although this term appears in more than 1000 publications in PubMed, there is no clear definition of this disease state. The original report by Kirkpatrick et al. did not include proposals for the minimal duration or minimal number of recurrences in a defined time period [9]. Of note, some authors use the term “syndromic CMC” in the context of associated autoimmunity, whereas others accept it as an isolated infection-based entity [9,10,11].

The *C. albicans* is the most common isolate causing CMC. When facing a patient with CMC, the physician should review and consider risk factors often associated with this disease, such as: concomitant antibiotic use, topical or systemic corticosteroid therapy, diabetes mellitus, secondary immunosuppression (e.g., chemotherapy and/or radiotherapy for hemato-oncologic diseases), HIV, as well as underlying (congenital) alterations of the immune system [8,12].

In the last few years, several IEI have been associated with CMC, many of which present in childhood. The study of these rare diseases has provided us with a deeper understanding of the pathophysiologic mechanisms involved. Overall, alterations and imbalances of IL-17 and IL-22, and possibly IFN-γ, have been identified as important factors predisposing individuals to develop CMC [11,13,14,15,16,17,18] Figure 1.

The CMC may be the presenting symptom in patients with inherited T cell deficiencies presenting as severe combined immunodeficiencies ((S)CID). This group is fairly heterogeneous and includes various subtypes that differ in their clinical manifestations and severity, laboratory findings, causal genes, and management [19]. Generally, patients with (S)CID are susceptible to a broad range of infectious agents. Pneumonia from *Pneumocystis jiroveci*, another fungal infection, is pathognomonic for T cell deficiencies, including (S)CID, and can be the life-threatening initial presentation [20]. The T cells in (S)CID patients are deficient in numbers and/or function [1,21]. In addition, some of these patients may be detected by systematic neonatal screening programs quantifying T cell receptor excision circles (TRECs) in neonatal dried blood spots [22], whereas for others, the diagnosis will be established later, through abnormal lymphocyte subsets and/or immunoglobulin results. The genetic testing for disease-causing mutations in underlying genes will help to definitively establish the molecular diagnosis [19]. The management of these patients differs according to the clinical presentation, the complementary laboratory results, and the affected gene, but often includes infection prophylaxis (isolation, antibiotic and/or antifungal prophylaxis, immunoglobulin replacement), and supportive care (e.g., nutrition) [23]. The early evaluation of curative treatment options such as hematopoietic stem cell transplant (HSCT) or, in selected cases, gene therapy, is necessary [19].

In contrast to patients with (S)CID, other IEI marked by CMC can be associated with discrete syndromes (including the Hyper-IgE syndrome [HIES], or autoimmune polyendocrinopathy-candidiasis-ectodermal dystrophy [APECED]); susceptible to other prototypical infections (e.g., *S. aureus* or mycobacteria); or occur in isolation.

The dominant-negative (DN) STAT3 mutations are responsible for autosomal dominant (AD) HIES, or Job’s syndrome, which is a complex disorder with hematopoietic and non-hematopoietic clinical manifestations. The classical clinical phenotype includes early-onset rash and eczema, bone fractures, delayed dentition, “cold” skin abscesses (due to *S. aureus*), recurrent sinopulmonary infections with pneumatocele development, and characteristic facial features [24,25]. In addition, the elevated eosinophil counts and IgE levels in blood are characteristic laboratory features for HIES, but they are not always present, or may be present intermittently. The CMC is a key manifestation of HIES due to DN-STAT3 mutations. Further, the invasive *Candida* infection is rare, but patients with previous lung damage are at risk of invasive aspergillosis (see section below). Impaired Th17 differentiation with decreased proportions of IL-17- and IL-22-producing T cells is likely responsible for the increased susceptibility to mucocutaneous *Candida* and *S aureus* infections in DN-STAT3 HIES [26,27,28]. The management of this IEI consists of antibiotic prophylaxis, supportive care, and antifungal prophylaxis when lung damage has occurred. The early treatment of potential infections is recommended, as many patients may have important bacterial infections without displaying significant inflammatory signs [24,29]. Further, hematopoietic stem cell transplantation has been performed to date in a small group of patients, restoring some immunologic alterations. However, non-hematological complications such as vasculopathy or bone-related complications will most likely not benefit from this procedure [29,30,31].

The CMC may be seen in other causes of HIES. For example, patients with autosomal recessive (AR) mutations in *ZNF341* [32,33] and *PGM3* [34,35] show disease manifestations resembling DN-STAT3 HIES, including CMC. The ZNF341 is a transcription factor that binds to the STAT3 promoter. In addition, the biallelic mutations in *ZNF341* lead to the loss of its function, decreasing STAT3 production and, thus, its function. Further, the ZNF341 deficiency is managed similarly to DN-STAT3 HIES. PGM3 is a congenital disorder of glycosylation and has been occasionally reported in association with CMC [34]. The exact mechanism by which biallelic mutations in PGM3 increase susceptibility to CMC is not clear, although slightly decreased Th17 levels have been described [34].

The CMC is one of the most common presenting symptoms in children with autoimmune polyglandular syndrome type 1 (APS-1, also named APECED; OMIM 240300): 25–50% of affected patients present with the CMC in the first year of life, with rates reaching 80–90% in the adult population [36,37,38]. The APECED is a rare (1:100,000) monogenic IEI due to mutations in *AIRE*, that are classically autosomal-recessive (AR), although dominant-negative variants have also been reported [39]. The typical APECED patients present with CMC, hypoparathyroidism, and primary adrenal insufficiency, although other autoimmune manifestations, such as pneumonitis and enteropathy, as well as enamel hypoplasia, have been described. In APECED, loss of AIRE function results in thymic dysfunction with the escape of autoreactive T cells. The lymphocytic organ infiltration, in combination with the generation of anti-cytokine autoantibodies, causes the most characteristic disease manifestations. The autoantibodies to IL-17 have been traditionally associated with CMC [40,41]. However, recent data focusing on gingival tissue suggest a more complex interaction beyond circulating T cells and include impaired type 17 mucosal immunity as well as immunopathology promoted by excessive type 1 mucosal inflammation [17,18]. The contribution of the latter mechanism may be supported by the therapeutic effect of targeted treatment strategies such as JAK inhibitors (JAKinibs) [17].

The CMC may also be due to IEI without one of the above syndromes, but in association with increased susceptibility to other infections. However, the DN-STAT3 HIES demonstrated the potential non-redundancy of the IL-17 pathway in susceptibility to CMC and *S. aureus* skin and soft tissue infection (SSTI). The discovery of AD IL-17F and AR IL-17RA deficiencies underlying CMC pinpointed the critical nature of this cytokine in human immunity to *Candida* [42]. In addition to CMC, patients may also develop staphylococcal SSTI. Consistent with this pathophysiological paradigm, AR ACT1 deficiency is another IEI that can manifest with CMC and *S. aureus* SSTI [43,44]. ACT1 is an adapter protein recruited to the IL-17 receptor, where it binds the IL17RA subunit, and mediates downstream signaling [45]. The biallelic mutations in the gene encoding ACT1, *TRAF3IP2*, result in impaired NF-κB activation. In a case series of published reports (n = 12) [46], CMC occurred in early childhood (before age 2 years) in 80% of cases, while *S. aureus* SSTI were documented in about 50% of cases. The standard immune phenotyping and immunoglobulin levels were unremarkable in most patients. However, the treatment responses of CMC were satisfactory when documented, and no fatal cases have been described [46]. The AD JNK1 deficiency has only been reported in three individuals from one family [47]. The JNK1 protein is part of the IL-17 and TGFβ1 signaling pathways. TGFβ1 is involved in the Th17 differentiation process, and its compromise due to mutant JNK1 likely explains the reduced proportion of ex vivo and in vitro differentiated Th17 cells found in all patients. Similar to the aforementioned etiologies, the patients’ clinical phenotype involved early-onset CMC and *S. aureus* SSTI. In addition, all patients had features suggestive of an Ehlers-Danlos such as connective tissue disorder, most likely due to the abnormal TGFβ1 signaling [47].

The susceptibility to CMC can also occur in association with ‘intracellular pathogens,’ notably mycobacteria. In patients with autosomal recessive IL-12p40 and IL-12RB1 deficiency, CMC can occur in ~25% of patients, although it is not necessarily concomitant with other infections [48,49]. Similarly, patients with autosomal recessive RORγt deficiency, due to biallelic mutations in *RORC* (encoding the RORγt transcription factor fundamental for regulating Th17 development), have susceptibility to both CMC as well as non-tuberculous mycobacterial infections [50]. These IEI can also predispose to invasive disease with other fungi (see below).

The CARD9 deficiency is the only known IEI that predisposes to both CMC and invasive candidiasis. The CARD9 encodes an adaptor protein associated with multiple C-type lectin receptors (CLRs), such as Dectin-1, which are involved in the recognition of fungus and subsequent pro-inflammatory response [51,52]. The loss of CARD9 function leads to variably diminished, but not abrogated, Th17 responses, potentially contributing to occasional CMC [53,54,55,56]. More strikingly, CARD9 deficiency leads to spontaneous development of invasive candidiasis and, distinctly, to central nervous system (CNS) involvement (see the Section 2.1.2 below).

The most common IEI in the context of CMC are STAT1 gain of function (GOF) mutations [57,58,59]. In some CMC cohorts, about half of the cases were diagnosed with this disease [57,60]. In a large international cohort of patients with such features, the search for underlying gene defects using a targeted sequencing approach yielded a diagnosis in 37.5% (24/64) of those with CMC, including: STAT1 GOF (63%), CARD9 (17%), STAT3 (8%), IL17RA (8%), and AIRE (4%) [61]. Of note, these results were obtained from a cohort that included many patients from Middle Eastern countries. Consanguinity is more common and may favor AR disorders, whereas cohorts with patients of European ancestry would likely reveal higher proportions of STAT1 and STAT3 defects [25,61].

Although the precise pathophysiological mechanisms by which STAT1 GOF predisposes to CMC still need to be elucidated, it appears that at least one determinant of disease is that increased activation of the JAK-STAT1 pathway results in unbalanced Th17 differentiation [62,63,64]. The patients with an autosomal dominant (AD) STAT1 GOF mutation present most commonly with early onset (first 2 years of life) CMC. Other disease manifestations, such as recurrent (myco-) bacterial, viral, and non-*Candida* fungal infections, have also been reported. Although patients may show reduced numbers of T cells and hypogammaglobulinemia, it is also not uncommon that standard immunologic evaluations with lymphocyte subsets (including Th17), immunoglobulin levels, and vaccine responses are normal. Therefore, genetic testing should be pursued early in such an evaluation. The significant (multiorgan) autoimmune manifestations, vascular abnormalities (aneurysms), and an increased risk of malignancies (squamous cancer) are also part of the broad clinical phenotype. The management of these patients is challenging as it often requires the combination of immunosuppression as well as anti-infective therapy [24,60]. In addition, hematopoietic stem cell transplantation (HSCT) is the only curative treatment option, but the current literature indicates high rates of secondary graft failure and mortality [65,66]. The JAKinibs such as ruxolitinib or baricitinib, have been shown to effectively treat many of the aforementioned disease manifestations, and in particular CMC [67,68,69,70,71]. A recent study summarized the experience with JAKinibs in pediatric STAT1 GOF patients and showed a good response rate (82%; 18/22 subjects) after 1–8 weeks of therapy. Further, most patients were able to discontinue previously prescribed antifungal prophylaxis [72]. It is noteworthy that, although JAK inhibition has now been used in a considerable number of patients, no guidelines exist regarding optimal dosing, monitoring, or follow-up. However, the long-term effects of JAK inhibition in STAT1 GOF, especially in the pediatric population, are yet unknown. Recently, a multinational consortium under the umbrella of ESID/IEWP and ERN has started to elaborate a consensus guideline aiming to address the aforementioned uncertainties [73].

In distinction to the above IEI, isolated CMC (to date) has been described in 3 subjects from different families with AR-complete IL-17RC deficiency [74]. Whether this IEI represents a finite CMC susceptibility, a phenotype in progress that will be revealed with time, or an additional reported case, it cannot be addressed currently.

The CMC management system is not standardized. Overall, acute and infrequently recurring (<2x/year) OPC episodes can likely be successfully treated with oral fluconazole for 3–4 weeks if congruent with antifungal susceptibility testing of isolates. In the case of azole resistance, echinocandins may be an appropriate alternative, although they currently require intravenous administration. Patients with frequently recurring (>3x/year) or persistent CMC should receive secondary prophylaxis with either triazoles or an oral cochleated amphotericin solution, which shows promise and may become an alternative option [75]. It should be noted that inadequately treated CMC may have important sequelae: For both APECED and STAT1 GOF, CMC most commonly affects the oral and esophageal mucosa and may lead to esophageal strictures and stenosis, while in some cases, squamous cell cancer has been reported as a long-term complication [36,60,76,77].

#### 2.1.2. Invasive Candidiasis (IC)

The IC is a growing health care problem and is considered the most common fungal disease among hospitalized patients in the developed world [3]. Candidemia, chronic disseminated candidiasis (previously known as hepatosplenic candidiasis), and CNS disease (e.g., meningitis) are often life-threatening and associated with important sequelae. Early diagnosis and prevention are key to avoiding deleterious complications. Risk factors in children include prematurity, damage to the gastrointestinal epithelial or skin (e.g., surgery, indwelling catheter, chemotherapy-associated mucositis, alteration of the microbiota due to the use of broad-spectrum antibiotics), as well as pharmacologic immunosuppression (e.g., corticosteroids or chemotherapy), and of particular interest for this review, a limited number of IEI [8,56].

In a patient with IC but lacking the above-mentioned iatrogenic risk factors, an underlying IEI should be considered and investigated. Specifically, IEI with alterations in the number or function of phagocytes should be ruled out. In this regard, IC has been reported in patients with congenital neutropenia syndromes (ELANE, HAX1, etc.) [78] and leukocyte adhesion disorders 1 (LAD-1, ITKB2) [79]. Similarly, complete myeloperoxidase (MPO) deficiency or chronic granulomatous disease (CGD) have been associated with deep-seated *Candida* infections [80,81,82]. The defective production of reactive oxygen species, which is required for an effective oxidative burst permitting elimination of *Candida* and other stereotypical microorganisms (e.g., *Aspergillus*; specific bacteria), is the most likely responsible pathophysiologic mechanism [82,83]. Of note, the rates of *C. lusitaniae*, a fairly uncommon *Candida* species, are substantially higher in CGD patients, potentially indicating a specific relevance of oxidative burst for this pathogen [84,85].

As stated above, patients with AR CARD9 deficiency are at risk for superficial and, more strikingly, invasive candidiasis (as well as other fungal infections). Importantly, these infections can manifest at any age [16,56,86,87]. The IC may affect various organs such as the bones, the gastrointestinal tract, and the eyes. CNS candidiasis (or meningoencephalitis) is, however, the most characteristic disease manifestation for patients with CARD9 deficiency. Therefore, children as well as adults presenting with spontaneous CNS candidiasis (e.g., meningitis, abscess), in the absence of obvious risk factors such as intraventricular shunts or head surgery, should be evaluated for this IEI [14,56,86,88]. The reason for this increased predilection of *Candida* infections to the CNS in patients with CARD9 deficiency has not been elucidated yet. However, intriguingly, CNS candidiasis is often associated with a mitigated neutrophilic response (tissue neutropenia). Candidalysin (a cytolytic peptide toxin produced by various *Candida* spp.) induces local microglia to produce interleukin IL1β and the C-X-C motif chemokine in a CARD9-dependent manner, enabling the recruitment of neutrophils to the CNS [89,90]. In addition, CARD9-deficient neutrophils have a diminished capacity to kill unopsonized yeast [91]. Although a number of patients have been described as having eosinophilia, raised IgE levels, or even a CVID-like phenotype [92], normal results in immunologic evaluations do not exclude this diagnosis, and thus genetic studies are necessary to establish a definitive diagnosis. Treatment is challenging and consists of intensive and prolonged (sometimes life-long) antifungal therapy. HSCT has been successfully performed in some patients [93].

### 2.2. Aspergillosis

*Aspergillus* spp. accounts for the majority of inhaled mould infections (IMI) [94]. Although *A. fumigatus* is the most common cause of aspergillosis, it has been recently suggested that the identification of emergent and rare *Aspergillus* species may indicate an underlying IEI [95,96]. The diagnosis of IA in patients without underlying risk factors, such as pronounced or prolonged immunosuppression or abnormal lung parenchyma, should trigger the investigation for IEI. Neutrophils are the most relevant immune cells involved in protection from IA. Therefore, syndromes associated with reduced numbers (e.g., LAD1 deficiency) or alteration of its function (e.g., CGD) should be suspected. Similar to patients without IEI, pulmonary aspergillosis is the most common manifestation, whereas extrapulmonary aspergillosis is rare and has only been reported in some patients with IEI [97,98].

#### Pulmonary Aspergillosis

CGD is by far the most common IEI associated with IA. *Aspergillus fumigatus* is usually the most common cause of disease, whereas identification of the otherwise lowly pathogenic *A. nidulans* is very specific for CGD patients [82,99]. The distinct susceptibility to *A. nidulans* in CGD may relate to loss of NADPH-oxidase-dependent NETosis [100]. Overall, the risk of a CGD patient developing IA at some stage of the disease ranges from 25 to 45% [95,101]. The first step in evaluating a patient with suspected CGD is usually the determination of the oxidative burst capacity, ideally by dihydrorhodamine (DHR)-based flow cytometry. Low or absent levels are highly suggestive of CGD. In addition, very low levels have been shown to translate into higher risks to developing IA. In addition to the absolute level of production, the histogram distribution of the oxidative burst responses by neutrophils is also important, as a modest diminution of the DHR response with a broad-based histogram, suggestive of autosomal recessive CGD, may not be identified purely by absolute quantification reporting. The genetic testing with molecular confirmation will finally establish the definitive diagnosis [81,82,83].

Patients with a high suspicion or established diagnosis of CGD should receive lifelong primary or secondary antifungal prophylaxis with drugs covering *Aspergillus*, such as itraconazole or posaconazole, whereas the role of prophylactic IFNγ remains under debate [102,103,104]. It is noteworthy that the serum galactomannan assay, often employed as a diagnostic surrogate test for IA, is not clinically reliable for the diagnosis of IA in the setting of CGD [105,106,107]. However, invasive procedures, such as bronchoscopy or tissue biopsies, are often necessary and may need to be repeated to identify the causal pathogen. The tissue samples should be submitted to microbiology for culture and, if available, molecular testing, as well as histopathologic evaluation [107]. A precise diagnosis of the responsible fungus is fundamental to developing an optimal management plan.

The patients with DN-STAT3 HIES (see above) are also at risk to develop pulmonary aspergillosis. These infections are principally found in patients with already established structural lung damage (e.g., pneumatoceles, cavities) secondary to previous bacterial infections [108]. As a result, several authors recommend antimould prophylaxis only for those patients with documented lung tissue damage [25,29]. Furthermore, a recent cohort study (n = 74) from France suggested that IA might be less common. The study reported 21 episodes of pulmonary aspergillosis in 13 (17.5%) STAT3-deficient patients, as well as aspergillomas (n = 5), chronic cavitary pulmonary aspergillosis (n = 9), and different forms of allergic bronchopulmonary aspergillosis-like diseases (n = 7) [109].

The pulmonary aspergillosis has also been reported in other IEIs, albeit with a much lesser frequency. Although CMC is the most representative fungal infection in patients with STAT1 GOF (see above) mutations, about 10% develop invasive fungal infections. Pulmonary aspergillosis was reported in 5/274 (1.8%) of patients [60,110]. The GATA2 haploinsufficiency is a complex disorder predisposing to a variety of infections [111,112,113,114,115,116,117]. Pulmonary aspergillosis has been reported in 6/124 (4.8%) patients [118]. Significantly, pulmonary alveolar proteinosis, which is an organ specific complication of GATA2 haploinsufficiency that occurs in about 10% of patients, is associated with higher rates of pulmonary aspergillosis (36%); see [119]. The role of antimould prophylaxis in these patients is, however, not clear.

### 2.3. Extrapulmonary Aspergillosis (EPA)

There are only a very limited number of IEIs in which EPA has been reported. AR CARD9 deficiency has been reported in rare cases associated with spontaneous CNS aspergillosis, intra-abdominal aspergillosis, and chronic cutaneous aspergillosis [97,98]. Meanwhile, AD-HIES due to DN-STAT3 mutations (see above) has been reported in association with vertebral [120] or sino-orbital aspergillosis [121].

#### 2.3.1. Thermally-Dimorphic Endemic Mycoses

The thermally dimorphic endemic fungi (TDEF) include *Blastomyces*, *Coccidioides*, *Emergomyces*, *Histoplasma*, *Paracoccidioides*, and *Talaromyces*. Each has its own classical geographic distribution; however, with the development of technologies facilitating detection of fastidious fungi in environmental and animal sources, improved diagnostic tests facilitating detection in humans, as well as climate change, the boundaries of these fungi are in flux. In addition, each of these genera has an increasing number of species, and some species have diverse clades/strains, which may account for some differences in clinical manifestations. Nonetheless, the overall pathophysiology of the TDEF involves a mycelial form in the environment, with small hyphal fragments and conidia constituting the infectious propagules that become inhaled, and consequently, the lungs are the main portal of entry. The relatively low prevalence of these fungal diseases in endemic areas implies that otherwise healthy individuals are able to contain that exposure; frequent but asymptomatic exposure is, itself, supported by screening programs based on seroprevalence or rates of delayed hypersensitivity testing. The following IEI are associated with increased susceptibility to disease from these fungi: Mendelian Susceptibility to Mycobacterial Disease (MSMD); GATA2 deficiency; CD40 ligand (CD40LG) deficiency; and autosomal dominant hyper-IgE (Job’s) syndrome (AD-HIES). Furthermore, these same IEI confer increased susceptibility to cryptococcosis. The *Cryptococcus* is polymorphic, consisting of a vegetative yeast phase capable of budding and hyphal growth during the sexual cycle; desiccated yeast and the spores resulting from hyphal mating act as infectious propagules to enter the host’s lungs. While not considered a TDEF, its biology and, more importantly, its immunopathology with respect to IEI overlap with those of the TDEF.

The MSMD refers to a group of disorders marked by impaired production of or response to interferon gamma (IFN-γ) [Table] [122,123,124]. In addition, the IFN-γ is primarily secreted by activated T cells and natural killer (NK) cells, to promote macrophages activation [Figure 2]: Following pathogen phagocytosis, macrophages secrete interleukin (IL)-12 and IL-23. The IL-12 is a heterodimer of p35 and p40; the latter IL12p40 subunit can also dimerize with p19 to form IL-23. IL-12 and -23 act on their cognate receptors on NK and T lymphocytes, resulting in the production of IFN-γ that acts on its cognate receptors on mononuclear cells (e.g., tissue macrophages, recruited monocytes, dendritic cells) to enhance intracellular killing, or at least contain the potential pathogen. Furthermore, the MSMD was initially described in the context of increased susceptibility to mycobacteria, especially non-tuberculous mycobacterial disease that is extra-pulmonary. A significant proportion (~30%) of MSMD patients can develop superficial (mucocutaneous) candidiasis (discussed above) [48,49,125]. However, MSMD may also present with otherwise-unexplained susceptibility to invasive disease due to TDEF (*Histoplasma* [48,126], *Coccidioides* [127,128], *Paracoccidioides* [129], *Cryptococcus* [130,131], either without or with a previous history of mycobacterial infection.

As listed in Table 1, mutations in *STAT1* can cause MSMD. More specifically, biallelic loss-of-function (LOF) mutations in *STAT1* cause an autosomal recessive syndrome with either complete or partial deficiency of STAT1, causing susceptibility to mycobacteria and viruses that can be either severe (life-thr eatening) or non-severe, respectively [122,123,125]. Additionally, heterozygous LOF mutations in *STAT1* can cause an autosomal dominant MSMD, through negative dominance [132,133,134]. In contrast to these LOF mutations causing MSMD, there are also heterozygous mutations that are gain-of-function (GOF) in STAT1 that are not considered genetic etiologies of MSMD *per se* but that do underlie susceptibility to TDEF or *Cryptococcus* by compromising the dynamics of the IFN-γ response [135,136]. The distinction between LOF and GOF in STAT1 requires experimental proof. While this distinction may be esoteric for non-IEI clinicians, the key point is that identification of variants in STAT1 may underlie unexplained severe fungal disease in patients, and further consultation with experts in the field may be required for a mechanistic interpretation of the variant and its relevance to clinical management.

The mycoses associated with defects of the IFN-γ pathway vary in age of onset and tend to be extrapulmonary (e.g., involvement of the lymph node, bone/bone marrow, skin, and CNS), recurrent, and/or severe. The severity of the clinical manifestation may depend on the underlying gene lesioned, although there are too few mycoses cases to date to robustly support that premise. The treatment with antifungals alone or combined antifungal/IFN-γ has been variably successful. In addition, the HSCT may be an option, although its experience as a treatment modality for refractory fungal disease is limited. Based on the experience with mycobacterial disease, HSCT could be considered the only curative treatment option, although delayed engraftment or graft failure, inextricably linked to elevated endogenous levels of IFN-γ in some forms of MSMD, may occur [137,138].

The GATA2 deficiency is a monogenic bone marrow failure syndrome that can manifest as immunodeficiency to mycobacterial, fungal, and/or viral disease (especially HPV-related warts) and increased susceptibility to hematologic malignancy (MonoMAC) [111,112,139], but also as dendritic cell, monocyte, B, and NK lymphoid (DCML) deficiency [113], familial acute myeloid leukemia [140], lymphedema with myelodysplasia/leukemia (Emberger syndrome) [141] or warts (WILD syndrome) [115], chronic neutropenia [142], and classical NK cell [143]. GATA2 is a transcription factor and master regulator of hematopoiesis, and mutations causing haploinsufficiency underlie the molecular basis for disease [116]. Notably, only one-third of known mutations are inherited (i.e., of parental origin), whereas the remaining two-thirds of cases are due to de novo mutations, highlighting that the absence of a family history of disease does not exclude this possibility [116]. Moreover, the clinical penetrance of the polymorphic manifestations of the possible syndromes is quite variable, depending on the phenotype considered. Overall, the clinical penetrance for any individual GATA2 deficiency-related disease phenotype was incomplete (32.9%) by age 40 [116], underscoring the fact that infections and other disease-specific features can manifest later in adulthood, despite the genetic nature of the syndrome. Among the immunologic defects in GATA2 deficiency, progressive monocytopenia is accompanied by varying degrees of B- and NK-lymphopenia, T-lymphopenia, neutropenia, or dendritic cytopenia, which may increase with age.

The GATA2 deficiency has been associated with disseminated histoplasmosis [111], coccidioidomycosis [144], and cryptococcal meningitis [111], as well as a single case of cavitary pulmonary blastomycosis [145]. The molecular basis for the susceptibility to TDEF in GATA2 deficiency has not been defined, but given the overlap with MSMD (above) and other IEI (below), it likely involves dysfunction of the IL-12/IFN-γ axis, perhaps through monocytopenia and/or macrophage-intrinsic defects. The response to antifungal therapy has been variable, although this may coincide with the development of other features of GATA2 deficiency, notably hematologic malignancy, which may complicate treatment. Given the proclivity of the disease to evolve into a myelodysplastic syndrome (MDS) and subsequent progression to hematologic malignancy (most commonly, acute myelogenous leukemia or chronic myelomonocytic leukemia), curative therapy with allogeneic hematopoietic stem cell transplantation should be considered early and ideally, before potential damage to the lung occurs (e.g., pulmonary alveolar proteinosis). This usually corrects the infection susceptibility as well, although the optimal timing of the transplant may be difficult to determine [146,147,148,149].

The loss of function mutations in CD40 ligand (*CD40LG*, encoding the protein CD40L) underlie X-linked hyper-IgM syndrome (XL-HIGM), characterized by the sentinel phenotype of elevated serum IgM levels, hypogammaglobulinemia of other antibody classes, and recurrent respiratory tract infections in young males [150]. This presentation is due to the fact that, CD40L normally expressed on activated CD4+ T cells, interacts with CD40 on B cells, leading to class switch recombination and allowing the transition from the IgM-class of antibodies to IgG, IgA, and/or IgE. However, CD40L also binds CD40 on macrophages and dendritic cells, and loss of this interaction causes defective cell-mediated immunity, resulting in a combined immunodeficiency clinical phenotype [150,151,152,153]. XL-HIGM increases susceptibility to various TDEF, including *Histoplasma*, *Talaromyces marneffei*, and *Paraccoidioides* [154,155,156,157]. All cases have been disseminated. XL-HIGM also predisposes to *Cryptococcus* [158,159,160,161,162,163,164]. In this context, cryptococcosis can present with ‘classical’ manifestations (i.e., involvement of the CNS) as well as ‘atypical’ presentations, including lymphonodular, cutaneous, or visceral involvement.

The AD-HIES (Job’s syndrome) is due to loss-of-function (LOF), dominant-negative mutations in STAT3 [29] (described above). This syndrome can be marked by disease due to TDEF or *Cryptococcus* [108,165]. In distinction to the above presentations in other IEIs, histoplasmosis in STAT3-LOF syndrome can be bizarre and include upper airway involvement (e.g., laryngeal histoplasmosis; tongue ulcer), diffuse gastrointestinal involvement, as well as disseminated/visceral disease [108,165]. Coccidioidomycosis can present with CNS disease, often with concomitant lung disease, while cryptococcosis tends to be localized to the CNS and gastrointestinal tract [165]. One case of progressive disseminated talaromycosis (blood, bone marrow, and hepatosplenomegaly) has also been reported [166].

While the above IEIs constitute the key ones consistently associated with susceptibility to TDEF and *Cryptococcus*, individual reports of TDEF infections have been occasionally reported with other genetically-confirmed IEIs, such as histoplasmosis in nuclear factor kappa B (NF-kB) essential modulator (NEMO) deficiency or in DOCK8 deficiency [167] and mediastinal coccidiomycosis in cytidine nucleotide triphosphate synthetase 1 (CTPS1) deficiency [168]. Further reports of these associations are required to validate the consistency of the fungal susceptibility, along with mechanistic insights into the causal role of the lesioned gene in TDEF immunity. The TDEF and *Cryptococcus* infections have also been reported in clinical syndromes that were not genetically identified or investigated, including idiopathic CD4+ lymphocytopenia (ICL) [169,170,171,172,173,174,175,176,177,178,179,180] and common variable immunodeficiency (CVID) [181,182]. In these cases, the absence of a molecular diagnosis precludes a refined understanding of immunopathogenesis.

#### 2.3.2. Deep Dermatophytosis

Dermatophytes are filamentous fungi that are uniquely keratinophilic, whereby they obtain nutrients from keratin-rich tissues, thus causing superficial infections of the skin, hair, and/or nails [183]. Classically, dermatophytes were classified into three genera: *Trichophyton*, *Epidermophyton*, and *Microsporum*. More recently, genomic-based data have revised the taxonomy into seven genera: *Trichophyton*, *Epidermophyton*, and *Microsporum*, as well as *Paraphyton*, *Lophophyton*, *Arthroderma*, and *Nannizzia* [184,185]. Collectively, these fungi are further categorized by ecological niche into those that are anthrophilic (spread between humans), zoophilic (from animals), or geophilic (from soil). Although dermatophytes usually cause superficial infections, it has long been recognized that, in some individuals, they cause progressive, refractory infections extending deeper than the skin/hair/nails. This latter syndrome has been called by many terms, including tinea profunda or deep dermatophytosis (DD) [186]. The DD is often clinically resistant to treatment, even though there is no evidence that this is due to microbiological resistance (based on in vitro antifungal susceptibility testing), highlighting the contribution of immunological failure in pathogenesis. In addition, DD has been observed in persons with iatrogenic immunosuppression, but there have been historic accounts of striking disease in those without. In 2013, the first evidence that a monogenic defect of immunity caused DD was discovered: autosomal recessive CARD9 deficiency [187]. This finding has been replicated in other cases of DD globally, signifying that this genetic immunodeficiency is not restricted to niche regions [183,188,189,190,191,192,193,194,195,196]. As described above (in the section on candidiasis), CARD9 deficiency predisposes to other fungal diseases, but in a fascinating manner, not with increased susceptibility to infections by microbes from other kingdoms.

The DD has been primarily reported in adults, although several cases of the disease’s onset in childhood or adolescence have been described. In DD, fungal disease involves the skin in an extensive or diffuse distribution, occasionally appearing destructively (e.g., deep or large necrotic ulcers), with dissemination to the lymph nodes, bones, brain, lung, and/or other viscera. In addition, the biopsy of skin lesions typically reveals epidermal hyperplasia, hyphae beyond the cornified layer (as would be normally observed in immunocompetent individuals), and granulomatous inflammation. Further, the hyphae with granulomata are also noted in lymph nodes and visceral structures. Microbiologically, DD is overwhelmingly due to *Trichophyton* spp., esp. *T. rubrum*, which, in the absence of CARD9 deficiency, is the most prevalent dermatophytic cause of tinea corporis in adults [184]. The immunological basis by which loss of CARD9 function causes DD remains to be adequately deciphered. On the one hand, there is a paucity of neutrophil recruitment to sites of fungal invasion, indicating a tissue neutropenia, even though there is no circulating neutropenia [89,91,197]. In vitro, impaired Th17 responses have been demonstrated [53,86,198], although the exact mechanism by which this impairment results in selective susceptibility to DD is still under investigation.

While autosomal recessive CARD9 deficiency is the only IEI identified to date as underlying some cases of DD, not all patients with CARD9 deficiency develop DD, and typical, superficial dermatophytic infections alone may develop. In the latter case, standard dermatophyte-targeting antifungal therapy may be sufficient. The DD, however, tends to be refractory (in ~20%) or relapsing after treatment cessation (in ~2/3rd of cases), and different antifungals may need to be tried [93,188]. Given that CARD9 is primarily expressed by myeloid cells, allogeneic stem cell transplantation may be considered; in very limited experience, this modality appears to have been curative [93].

#### 2.3.3. Mucormycosis

Mucormycosis is a severe invasive disease due to fungi of the order *Mucorales*, of which the most common causes of human infections are *Rhizopus*, *Mucor*, and *Rhizomucor*. The most common syndromes are rhino-orbital-cerebral, pulmonary, cutaneous, gastro-intestinal, and disseminated disease. Mucormycosis is not classically stereotypical of any particular IEI or group of pathophysiologically linked IEI disorders; cases have only been sparsely and sporadically reported. Moreover, no specific mucormycosis syndrome or presentation is pathognomonic for an IEI except for the occurrence of deep disease in the absence of any immunocompromising state. Those IEI associated with cases of mucormycosis include CGD (especially in the context of steroid use) [199], STAT1 GOF [200], CARD9 deficiency [195,201,202], and GATA2 deficiency [139].

## 3. Conclusions

The presence of persistent, recurrent, or refractory superficial fungal disease or invasive disease of any kind in the absence of exogenous immunosuppression, especially in children, absolutely requires an investigation for an inborn error of immunity. The identified genetic variants can be cross-referenced with the existing literature to determine if they have been previously validated experimentally as a mutation causing an immune deficiency. If it is a novel variant, we strongly favour that mechanistic investigations be conducted to confirm that the variant is indeed deleterious to molecular and/or cellular function, with eventual publication, as the accumulation of these cases will expand our knowledge into genetic determinants of human susceptibility to fungal diseases.

## Figures and Tables

**Figure 1 jof-09-00149-f001:**
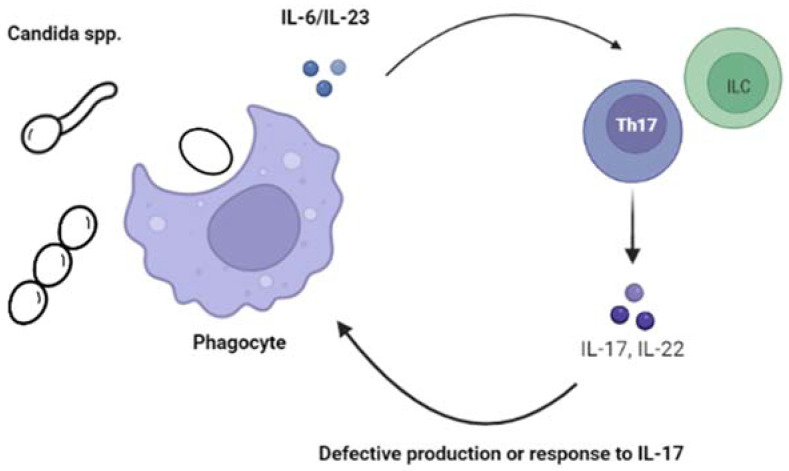
Alterations of the IL-17 immunity increases the susceptibility to chronic mucocutaneous candidiasis (CMC).

**Figure 2 jof-09-00149-f002:**
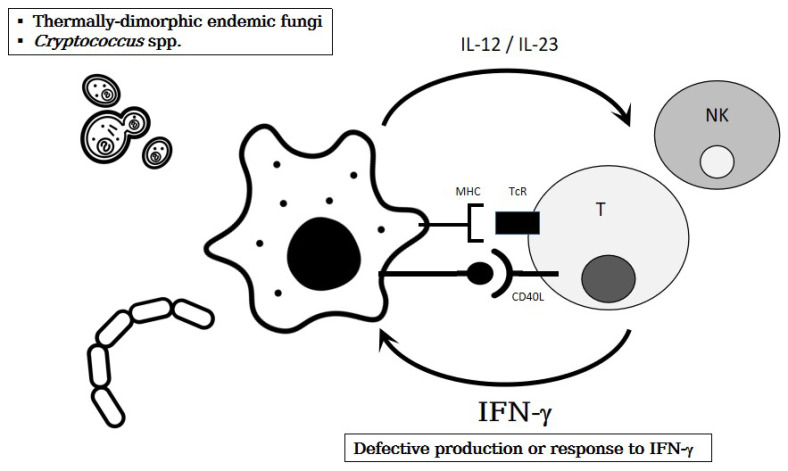
Impaired IFN-γ immunity increases susceptibility to mycoses from thermally-dimorphic endemic fungi and *Cryptococcus* spp.

**Table 1 jof-09-00149-t001:** The IEI discussed in this review are summarized.

Inborn Error of Immunity	Gene	Inheritance	Associated Fungal Diseases	Associated Clinical Features	Relevant Immunological Features	Comments
ACT1 deficiency	TRAF3IP2	AR	CMC	May also get *S. aureus* skin and soft tissue infections	Impaired signaling of IL-17 receptor	
APECED	AIRE	AR	CMC	Endocrinopathies (typically, parathyroid, adrenal, and gonadal insufficiencies)	Auto-antibodies to cytokines, including IL-17 and IL-22	Autoimmune polyendocrinopathy-candidiasis-ectodermal dystrophy; aka autoimmune polyglandular syndrome type 1 (APS1)
Autosomal dominant—Hyper-IgE syndrome	STAT3/Dominant-negative mutations	AD	CMC	Characteristic facial features. Musculoskeletal abnormalities (hyperextensibility; scoliosis). Decreased bone density/pathological fractures. Abnormal retention of primary teeth. Vascular anomalies. Ocular complications. Elevated serum IgE. “Cold” cutaneous *S. aureus* skin abscesses.	Impaired Th17 development	
			Pulmonary aspergillosis			Pulmonary aspergillosis & aspergillomas typically occurs in established structural lung damage (e.g., pneumatoceles, cysts, cavities)
			Aspergillomas, ABPA			
			Extra-pulmonary aspergillosis			
			TDEF			
			*Cryptococcus*			
CARD9 deficiency	CARD9	AR	CMC	Fungal disease may be adult in onset		
			Invasive Candidiasis			Especially, spontaneous CNS candidiasis
			Extrapulmonary aspergillosis			
			Deep Dermatophytosis			aka Tinea Profunda
			Mucormycosis			
Chronic Granulomatous Disease (CGD)	CYBB	XL	Invasive aspergillosis. May also get: Invasive candidiasis. Mucormycosis	Spontaneously occurring severe or recurrent bacterial infections of lung, liver, bone, skin, and/or lymph nodes. Granulomatous inflammation in gastrointestinal and/or genitourinary organs. Abnormal wound healing. Colitis, especially with fistulae and fissures.	Impaired phagocyte oxidative burst	Aspergillosis is the main fungal disease in CGD, primarily *A. fumigatus* complex and *A. nidulans* complex
	CYBA	AR				
	CYBC1	AR				Mucormycosis especially in the context of recent steroid use
	NCF1	AR				
	NCF2	AR				
	NCF4	AR				
CD40LG deficiency	CD40LG	XL	TDEF	Elevated or normal serum IgM. Decreased IgG, IgA, and IgE. Combined T and B immunodeficiency.	Impaired CD40L-CD40 interactions, leading to impaired costimulatory signal for T activation	X-linked Hyper-IgM syndrome (X-HIGM)
			*Cryptococcus*			
GATA2 deficiency	GATA2	AD	TDEF	Some features may be adult-onset. Susceptibility to mycobacterial and viral (esp. HPV) infections. May develop Pulmonary alveolar proteinosis.	Monocytopenia, B lymphopenia, NK lymphopenia. Variable T cell dysfunction.	Causes: MonoMAC, DCML, familial acute myeloid leukemia, Emberger syndrome, WILD syndrome, chronic neutropenia, and/or classical NK cell deficiency
			Pulmonary aspergillosis			Pulmonar aspergillosis especially in context of pulmonary alveolar proteinosis
			*Cryptococcus*			
			Mucormycosis			
IL17F deficiency	IL17F	AD	CMC	May also get *S. aureus* skin and soft tissue infections	Impaired IL-17 function	
IL17RA deficiency	IL17RA	AR	CMC	May also get *S. aureus* skin and soft tissue infections	Impaired IL-17 function	
IL17RC deficiency	IL17RC	AR	CMC		Impaired IL-17 function	Isolated CMC
JNK1 deficiency	JNK1	AD		May also get *S. aureus* skin and soft tissue infections	Impaired IL-17 and TGFβ1 signaling pathway	Features of Ehlers-Danlos like connective tissue disorder
MSMD						Mendelian Susceptibility to Mycobacterial Disease
	IL12B	AR	CMC	Susceptibility to Salmonella, Mycobacteria	Defect in production or response to IFN-γ	
	IL12RB1	AR	CMC	Susceptibility to Salmonella, Mycobacteria	Defect in production or response to IFN-γ	
			TDEF		Defect in production or response to IFN-γ	
			*Cryptococcus*		Defect in production or response to IFN-γ	
	IL12RB2	AR		Susceptibility to Salmonella, Mycobacteria	Defect in production or response to IFN-γ	
	IL23R	AR		Susceptibility to Salmonella, Mycobacteria	Defect in production or response to IFN-γ	
	TYK2	AR		Susceptibility to Mycobacteria.Isolated and syndromic forms exist. Syndromic form may have elevated IgE and susceptibility to viral diseases	Defect in production or response to IFN-γ	
	RORC	AR	CMC	Susceptibility to Mycobacteria	Defect in production or response to IFN-γ	
	IFNG	AR		Susceptibility to Mycobacteria	Defect in production or response to IFN-γ	
	IFNGR1	AD or AR	TDEF	Susceptibility to Mycobacteria	Defect in production or response to IFN-γ	
	IFNGR2	AD or AR		Susceptibility to Mycobacteria	Defect in production or response to IFN-γ	
	JAK1	AR		Susceptibility to Mycobacteria	Defect in production or response to IFN-γ	Impaired response to type I IFN
	STAT1/LOF/Negative dominance	AD		Susceptibility to Mycobacteria	Defect in production or response to IFN-γ	Impaired response to type I IFN
	STAT1/LOF	AR		Susceptibility to Mycobacteria	Defect in production or response to IFN-γ	Impaired response to type I IFN
	IRF8	AD		Susceptibility to Mycobacteria	Defect in production or response to IFN-γ	
	SPPL2a	AR		Susceptibility to Mycobacteria	Defect in production or response to IFN-γ	
	ISG15	AR		Susceptibility to Mycobacteria	Defect in production or response to IFN-γ	
	CYBB	XL		Susceptibility to Mycobacteria	Defect in production or response to IFN-γ	Defect in macrophage oxidative burst
	IKBKG	XL		Susceptibility to Mycobacteria	Defect in production or response to IFN-γ	
(Severe) Combined Immunodeficiency	IL2RG	XL	CMC		T- B+ NK- SCID	
	JAK3	AR	CMC			
	PTPRC	AR	CMC			aka CD45 deficiency
	CD3D	AR	CMC			
	CD3E	AR	CMC			
	CD3Z	AR	CMC			
	LAT	AR	CMC			
	IL7RA	AR	CMC		T- B+ NK+ SCID	
	CORO1A	AR	CMC			
	RMRP	AR	CMC	SCID with skeletal dysplasia, short stature, thin/sparse hair growth, neuronal colonic dysplasia (Hirschsprung-like anomaly), increased risk of malignancy		
	ADA	AR	CMC		T- B- NK-	
	AK2		CMC			
	RAG1/2	AR	CMC		T- B- NK+	
	DCLRE1C	AR	CMC			
	RAC2	AD/GOF	CMC			
	NHEJ1	AR	CMC	SCID with microcephaly, growth retardation	T- B- NK+	aka Cernunnos-XLF deficiency
	LIG4	AR	CMC			
	PRKDC	AR	CMC			
STAT1 GOF	STAT1/GOF	AD	CMC		Molecular gain of phosphorylation, with cellular loss of response to IFN restimulation due to prolonged phosphorylation. Impaired Th17 response	CMC may improve with JAK inhibitors
			TDEF			
			*Cryptococcus*			
			Mucomycosis			

## Data Availability

Data sharing not applicable No new data were created or analyzed in this study. Data sharing is not applicable to this article.

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
