# Peer review of "Inborn Errors of Immunity Causing Pediatric Susceptibility to Fungal Diseases"

_jof, 2023, doi:10.3390/jof9020149_

Round 1
Reviewer 1 Report
I found this review very useful sand well written I thisk there was a typo error: line 292 lunch instead of lung?
In general, if possible I would add some data regarding IFD in cancer patients and allo HSCT related with peculiar TLR mutations that can be neglected in the normal life but can become important in the case of induced immunosuppression. In my opinion it could add somethnig very useful to an already excellent review
Author Response
We thank Reviewer 1 for their strong rating of our manuscript (5 stars in all categories).
We have corrected line 292 accordingly.
The suggestion to include IFD in cancer patients and allogeneic HSCT with peculiar TLR polymorphisms is an excellent one, and certainly, the field of how common (>1% MSF) variants (SNP) in TLR genes may contribute to susceptibility to invasive fungal diseases in the context of chemotherapy/transplant is exciting. However, for this special issue on "Fungal Infections in Children 2022", we have been invited to specifically focus on "inborn errors of immunity", which is what we have done. The unmasked effects of TLR SNP by chemotherapy/transplant (or even biologic modifiers) are not, per se, inborn errors of immunity, which is why we have not included this topic. The special editor for this issue is Dr. Yael Shahor, and I am not sure if he has invited another author(s) to focus on IFD in chemotherapy/SCT patients; if so, this topic (TLR SNP) would probably be better suited for that manuscript.
Reviewer 2 Report
The present study entitled “Inborn Errors of Immunity causing Pediatric susceptibility to Fungal diseases” aimed to describe the advances in inborn errors of immunity associated with increased susceptibility to fungal disease”. Overall, the manuscript is well-written and presents an interesting state of the art, nevertheless, it still needs some points to be addressed, namely, a graphical abstract should be added to the manuscript to improve its readability and impact.
INTRODUCTION:
Line 47 – The Authors mentioned the Table at the final of the introduction section, however, it is missing the caption before the depicted table. Please review.
METHODS SECTION – Missing section
This is a crucial section that will improve the scientific value of the present review.
The author should add the Methods section after the Introduction. This is important to understand the search workflow and strategy of the present review.
Some point should be addressed, namely:
- Keywords
- Databases (for example: Pubmed and Web of Science, among others)
- Period of time
- Number of the studies
- Number of excluded studies
- Type of included studies (for example: larger cohort studies and case series)
- An overview of the included literature should be provided in a table, which could be a supplementary table.
I: Candidiasis – Page 8, Figure 1
III. Thermally-dimorphic endemic mycoses – Page 14, Figure 2
The Authors should be encouraged to improve the quality of the figure by including all the names of the structures represented. There are several free license programs that can be used for this purpose.
Author Response
A systematic search of the literature was performed in December 2022 to identify all publications reporting fungal disease in patients with IEI. Five databases were used: PubMed, Google Scholar, Web of Science, EMBASE, and Scopus. There was no restricted period of time; all articles dating back to the oldest one listed in the available database were identified. The literature search was performed using key words (and their derivatives) relevant to the fungi discussed in this review, including "fungi" ("fungal"), "yeast", "mould" ("mold"), "mycosis" ("mycoses"), "Candida" (and "candidiasis", "Torulopsis", species names), "Aspergillus" (and "aspergillosis", "Neosartorya", species names", "thermally-dimorphic", "endemic", "Histoplasma" (and "histoplasmosis", species names), "Blastomyces" (and "blastomycosis", species names), "Coccidioides" (and "coccidiodomycosis", "valley fever", "San Joaquin valley fever", species names), "Emergomyces" (and "emergomycosis", species names), "Paracoccidioides" (and "paracoccidioidomycosis", species names"), "Mucor" (and "mucormycosis", species names), "Rhizopus" (and species names), other fungal genera/species, in association with "inborn errors of immunity" (and "IEI"), "primary immunodeficiency" (and "PID", "PIDD"), "gene" (and "genetic", "mutation", "variant", "heterozygous", "homozygous", "loss of function", "LOF", "gain of function", "GOF", "dominant negative", "negative dominance", "haploinsufficiency", "haploinsufficient'), "deficiency", "immunodeficiency" (and "immune deficiency"), or individual IEI as listed in the "2022 IUIS Phenotypical classification for Human Inborn Errors of Immunity". The identified titles and abstracts were screened, and the full texts of suitable articles were reviewed.
Articles fulfilling the inclusion criteria were selected from the found full-text articles. To be eligible for inclusion, fungal disease must have been reported in patients from studies and case reports. Further, the IEI of patients must have been clinically specified and diagnosed by genetics. The relevant reviews and large case series were reviewed to identify potentially eligible original studies or unpublished data. We included all types of publications (articles, reviews, editorials, letters, and correspondences) written in English or French, and all references cited in these publications were also analyzed. Finally, where reports were identified that did not meet inclusion criteria, we have nonetheless included them with reference, but specifying that their contribution to the topic of this manuscript may be dubious.
Given the extensive list of references that we have included and cited in the text where appropriate, and the above search strategy explained in the new "Methods" section, we believe that this has strengthened the manuscript and it is not clear to us what the added value of providing an overview of the included literature as a table would be.
#2: The Authors should be encouraged to improve the quality of the figure by including all the names of the structures represented. There are several free license programs that can be used for this purpose.
Response: We have updated the figures to include the pertinent structures depicted. We have strived to find a right balance between fungal structures (given the intended readership for Journal of Fungi) and immunological structures (which graphically summarize what is detailed in the text). We hope that this satisfies the Reviewer.
The 2022 Update of IUIS Phenotypical Classification for Human Inborn Errors of Immunity.
Bousfiha A, Moundir A, Tangye SG, Picard C, Jeddane L, Al-Herz W, Rundles CC, Franco JL, Holland SM, Klein C, Morio T, Oksenhendler E, Puel A, Puck J, Seppänen MRJ, Somech R, Su HC, Sullivan KE, Torgerson TR, Meyts I. J Clin Immunol. 2022 Oct;42(7):1508-1520